# Understanding the Causes of Frailty Using a Life-Course Perspective: A Systematic Review

**DOI:** 10.3390/healthcare12010022

**Published:** 2023-12-21

**Authors:** Antonio Barrera, Leandro F. M. Rezende, Angelo Sabag, Christopher J. Keating, Juan Pablo Rey-Lopez

**Affiliations:** 1Faculty of Health Sciences, Universidad Internacional de Valencia (VIU), 46002 Valencia, Spain; abarsal@hotmail.com; 2Department of Preventive Medicine, Escola Paulista de Medicina, Universidade Federal de Sao Paulo, Sao Paulo 04023-900, SP, Brazil; leandro.rezende@unifesp.br; 3Discipline of Exercise and Sport Science, Faculty of Medicine and Health, The University of Sydney, Sydney, NSW 2006, Australia; angelo.sabag@sydney.edu.au; 4Facultad de Deporte, UCAM Universidad Catolica de Murcia, 30107 Murcia, Spain; cjames@ucam.edu

**Keywords:** frail, aging, prevention, bias, confounding

## Abstract

(1) Background: Few studies have examined risk factors of frailty during early life and mid-adulthood, which may be critical to prevent frailty and/or postpone it. The aim was to identify early life and adulthood risk factors associated with frailty. (2) Methods: A systematic review of cohort studies (of at least 10 years of follow-up), using the Preferred Reporting Items for Systematic Reviews and Meta-Analyses guidelines (PRISMA). A risk of confounding score was created by the authors for risk of bias assessment. Three databases were searched from inception until 1 January 2023 (Web of Science, Embase, PubMed). Inclusion criteria were any cohort study that evaluated associations between any risk factor and frailty. (3) Results: Overall, a total of 5765 articles were identified, with 33 meeting the inclusion criteria. Of the included studies, only 16 were categorized as having a low risk of confounding due to pre-existing diseases. The long-term risk of frailty was lower among individuals who were normal weight, physically active, consumed fruits and vegetables regularly, and refrained from tobacco smoking, excessive alcohol intake, and regular consumption of sugar or artificially sweetened drinks. (4) Conclusions: Frailty in older adults might be prevented or postponed with behaviors related to ideal cardiovascular health.

## 1. Introduction

Numerous potential biomarkers of aging have been proposed in the scientific literature, including molecular, imaging, and clinical data [1]. Frailty is a composite aging biomarker characterized by a condition of decreased physiological reserve that leads to a vulnerable state and increases the risk of adverse health outcomes when exposed to a stressor [1]. In 2001, two definitions of frailty were introduced in the geriatric literature (although more definitions can be found in the scientific literature). The phenotypic model of Fried et al. [2] is based on the presence of three (or more) of the following characteristics: (a) an involuntary weight loss, (b) self-reported exhaustion in daily life activities, (c) a low level of physical activity, (d) habitual slow walking speed, and (e) muscular weakness. Another frailty definition is the accumulation of deficits model (or frailty index) of Mitnitski et al. [3], which includes deficiencies of functional, sensory, and clinical nature. In a recent systematic review, the prevalence of frailty in older adults in 62 countries and territories was 12% (Fried phenotype) or 24% (frailty index) [4]. Women and individuals of a low socioeconomic status level are more likely to become frail, according to a narrative review of Taylor et al. [5]. Frailty remains an important public health problem because frail individuals (versus non-frail) are at higher risk of physical disability [2], falls [6], fractures [7], hospitalizations [8], institutionalization [9], and death [10].

In a recent systematic review of older adults (at least 65 years old) [11], authors identified a large number of lifestyle factors and characteristics associated with frailty. Information was mainly derived from cross-sectional studies or cohort studies with a short follow-up. However, it is well established that a life-course perspective offers a more suitable approach to understanding how the aging processes and their consequences emerge during the lifetime [12]. Lowering the accumulation of harmful exposures throughout the life course or changing unhealthy behaviors during adulthood may lead to more favorable trajectories of aging [12]. However, many statistical associations found in observational studies of risk factors of frailty could reflect bias (reverse causality, selection bias, and measurement errors), confounding, or chance [13]. To reduce the risk of bias due to pre-existing diseases in the synthesis of evidence, some epidemiologists recommend following some analytical approaches [14], such as (1) excluding (or adjusting for) participants with major noncommunicable diseases (NCDs) at baseline (CVDs, stroke, cancer, and respiratory diseases); (2) including only cohort studies with a minimum of 10 years of follow-up in meta-analysis; and (3) excluding death cases occurring in the first 5 years of follow-up.

To sum up, to the best of our knowledge, no systematic review until now has evaluated how early-life and middle-life risk factors are associated with frailty or studied their epidemiological validity using risk of bias assessments. The main objective of this systematic review was to identify early- and middle-life risk factors associated with frailty in older adults. We also aimed to examine whether authors included appropriate analytical methods to deal with confounding due to pre-existing diseases.

## 2. Materials and Methods

### 2.1. Literature Search and Screening

To formulate a clear and concise research question, a description of the population, intervention/exposure, comparison, and outcomes are provided. We searched cohort studies that examined associations between any risk factor (Exposure) during adulthood, adolescence, childhood, or natal factors (Population) associated with frailty (Outcome), using Web of Science, Embase, and PubMed (from inception until 1 January 2023). One researcher (A.S.) was in charge of producing the first database for the identification of relevant scientific literature. Details of the list of keywords are included in Appendix A. We followed the Preferred Reporting Items for Systematic Reviews and Meta-Analyses (PRISMA) 2020 guidelines to report the results of this systematic review [15]. Two authors, A.B. and J.P.R.-L., screened independently articles by title and abstract and, in a later stage, reading full-text articles using the website covidence.org In case of discrepancies, a third author (C.J.K.) made a final decision.

The eligibility criteria of this systematic review were any cohort study in humans (both sexes, healthy at baseline), published in English language, that evaluated associations of risk factors associated with frailty status in humans. Studies with a retrospective study design were eligible studies, as they inform about early-in-life risk factors. In addition, we considered eligible those studies cited in references from selected studies. Exclusion criteria: studies whose cohort studies had a follow-up lower than 10 years were excluded because they did not inform of early- or middle-life risk factors. Studies published in non-English language or abstracts of conferences were excluded.

### 2.2. Data Extraction

We retrieved the first author’s name, year of publication, country, sample size, participant’s sex, age at baseline, exposure variable/s, frailty definition, average length of follow-up, and fully adjusted hazard ratio (HR) or odds ratio (OR) or relative risk ratio (RR) and 95% confidence intervals (CIs) for frailty, comparing for each exposure variable (using 1-unit increase or comparations between categories with the highest and lowest values; Comparison). We also extracted data about the covariates used in the fully adjusted model and whether authors included sensitivity analyses in their publications. Data extraction was performed by one researcher (A.B.) and double-checked by another (J.P.R.-L.).

### 2.3. Risk of Confounding Due to Pre-Existing Diseases

Three methodological characteristics defined the risk of confounding based on subject matter expertise instead of a mechanistic risk of bias assessment [16]: average age at baseline of 70+ years, authors did not exclude participants with diseases/conditions at baseline and did not adjust for diseases/conditions in the fully adjusted model. A risk of confounding score was created with the three mentioned characteristics (ranging from zero to three). We defined a high risk of confounding due to pre-existing diseases when studies had two or three points in the confounding score (total of three). Scores were calculated by one senior researcher (J.P.R.-L.) with prior experience in risk-of-bias assessments of epidemiological studies. A comprehensive meta-analysis of each risk factor identified was initially planned, taking into account the risk of confounding scores, but it was finally discarded due to the scarce number of studies identified.

## 3. Results

Figure 1 shows the PRISMA 2020 flow diagram used in the present systematic review. A total of 7425 records were initially identified. After screening 5681 articles by title and abstract, 84 articles were retrieved for eligibility analyses through a full-text reading. Of these, 33 articles were finally selected [17,18,19,20,21,22,23,24,25,26,27,28,29,30,31,32,33,34,35,36,37,38,39,40,41,42,43,44,45,46,47,48,49]. Each section may be divided by subheadings. It should provide a concise and precise description of the experimental results, their interpretation, as well as the experimental conclusions that can be drawn.

Table 1 describes the main characteristics of all cohort studies selected. The population sample sizes ranged between 323 and 121,700 participants; 24 studies were conducted on both sexes, 4 only in men, and 5 only in women; 8 studies recruited participants in the USA, 7 in Finland, 1 in Australia, 4 in France, 6 in the United Kingdon, 1 in Israel, 1 in Sweden, 3 from China, and 1 in the Netherlands; participants were followed up between 10 and 30 years; the 11 definitions of frailty used across the cohort studies were Fried phenotype [2] (15 studies), Modified Fried phenotype (6 studies), FRAIL scale—Abellan van Kan et al. [50] (1 study), FRAIL scale—Morley et al. [51] (4 studies), Modified FRAIL scale—Morley et al. (1 study), Frailty index—Mitnitski et al. [3] (2 studies), Hospital Frailty Risk Score (HFRS)—Gilbert et al. [52] (1 study), Frailty phenotype—Kucharska-Newton et al. [53] (1 study), Frailty phenotype—Strawbridge et al. [54] (1 study), and Frailty index—Searle et al. [55] (1 study). The prevalence of frailty ranged between 2 and 61%.

The exposure variables of the 33 articles selected were dietary inflammatory index in adulthood (1 study), blood inflammatory markers in adulthood (two studies), alcohol consumption in adulthood (three studies), sitting time in adulthood (one study), multicomponent healthy heart score in adulthood (one study), overweight/obesity or higher BMI in adulthood (four studies), neighborhood-social deprivation in childhood (one study), cardiovascular risk scores in adulthood (two studies), physical inactivity in adulthood (five studies), asthma in adulthood (one study), anemia in adulthood (one study), diabetes in adulthood (one study), high liver enzymes (one study), dietary clusters (pasta or biscuits plus snacking) in adulthood (one study), birth body composition (BMI, weight, length) (one study), children of separated parents in childhood (one study), fruit and vegetable consumption in adulthood (three studies), smoking status in adulthood (two studies), nonsteroidal anti-inflammatory drugs (NSAID) use (one study), neighborhood quality in adulthood (one study), education level achievement (three studies), paternal education (one study), occupation or employment level in adulthood (two studies), low literacy in adulthood (one study), low income in adulthood (one study), malnutrition in adulthood (one study), depression in adulthood (one study), forced expiratory volume or HDL cholesterol or hypertension (one study), sugar-sweetened beverages or artificial sweetened beverage or orange juices or non-orange juices (one study), red meat in adulthood (one study), pain during walking in adulthood (one study), subjective social status (one study), health-related quality of life (one study), and social vulnerability index in adulthood (one study).

Table 2 shows the analytical approaches used to account for confounding due to pre-existing diseases in 33 studies examining the association between risk factors and frailty. In 10 of them, authors omitted the inclusion of any type of disease as a covariate in their fully adjusted models; only 2 studies excluded all participants with diseases in main analyses; and another 2 used sensitivity analyses.

Table 3 shows the scored risk of confounding due to pre-existing diseases in the 33 studies selected. The final score takes into account whether studies included participants younger than 70 years at baseline or not, whether studies excluded participants with diseases/conditions at baseline, and whether studies adjusted for diseases/conditions in the fully adjusted model. A total of 16 studies scored a low risk of confounding due to pre-existing diseases (0 or 1 point) and the rest a high risk of confounding (2 or 3 points).

Appendix A shows whether authors included physical activity or nutritional factors (energy intake, quality nutritional indexes, sugar-sweetened beverages, and red meat) as covariates in their regression models. In 20 studies, authors omitted both physical activity and nutritional factors as covariates in their statistical analyses.

## 4. Discussion

The goal of this systematic review was to identify early-life and middle-life risk factors (any exposure variable) associated with frailty. We found evidence that maintaining a normal weight in adulthood, being physically active, not smoking tobacco, refraining from ultra-processed food and beverages, and avoiding excessive alcohol intake may decrease the risk of frailty several decades later. These findings may have important implications for elderly populations because, in theory, most cases of frailty might be prevented if populations remain healthy before older age. For example, staying physically active in adulthood was robustly associated with a lower future risk of frailty [32]. The physiological mechanisms underlying the positive influence of physical activity on frailty prevention have been comprehensively reviewed elsewhere [5]. On the other hand, the regular consumption of fruits and vegetables and a low consumption of SSBs or ASBs were also associated with a lower risk of becoming frail [46]. Therefore, it seems unquestionable that diet and physical activity have a key role in preventing the future risk of frailty. In support of this, we found that obesity [21], diabetes [26], or having worse cardiovascular risk scores [27] or blood inflammatory markers [40] were equally associated with a higher risk of frailty. The mechanisms by which high ultra-processed foods and beverages promote obesity, diabetes, or cardiovascular disease are complex and only partially known nowadays. High ultra-processed foods and beverages may result in unique patterns of gut–brain signals during digestion processes, as they are absorbed more proximally in the gut compared with natural foods and beverages. Altered absorption of nutrients and a low amount of fiber in the diet may play important disruptions in appetite control, leading to a long-term positive energy balance [56].

Nonetheless, a second goal of our review was to evaluate whether authors included appropriate analytical methods to deal with confounding due to pre-existing diseases and we found serious deficiencies in this matter. For example, many (half of the studies selected) epidemiological studies of risk factors of frailty were at high risk of confounding due to pre-existing diseases. Another source of concern was the observation that the majority of studies did not adjust their effect estimates by important confounders such as physical activity or nutritional factors. The framing that frailty is a direct consequence of the normal aging process should be approached with caution, as we also find evidence that frailty was associated with worse socioeconomic markers (income, education, and employment). So far, the mechanisms linking socioeconomic factors with frailty remain unexplored, and multiple factors may be involved (beyond the proven benefits of physical activity or healthy diets). Although the best way to define frailty is still debated among scientists [51], future studies should adopt (at least) the most common definition of frailty (Fried phenotype) to allow a future synthesis of the scientific evidence [57].

To move the field forward, it is important to acknowledge that well-powered randomized clinical trials, although the gold standard of scientific inquiry, are limited in their ability to add valuable insights for prevention because it is unfeasible to test human interventions with a duration of 10 years or more. To illustrate how clinical trials in the elderly do not always offer important insights about how to prevent frailty, see reference [58], where a complex intervention that combined a nutrition plus physical activity intervention over 2 years was ineffective in reducing frailty in older adults. Although it could be argued that the design of interventions was not optimal (for example, short duration or the physical activity intervention only included 1 h per week of strength plus balance instead of exercise programs of aerobic activities based on physical activity recommendations for health in adults), the authors stressed that their interventions mirrored the real world (good external validity). Consequently, we think that future studies on this topic should rely on well-designed cohort studies with valid methodologies of assessment of exposure variables and robust statistical analysis (including sensitivity analyses). Our systematic review examines, for the first time, what modifiable factors may determine a higher long-term risk of frailty (life course epidemiology) and evaluates the risk of confounding due to pre-existing diseases. Despite employing a comprehensive search strategy, we found very few studies evaluating the same exposure variable, which precluded our ability to perform additional meta-analyses. Therefore, we acknowledge that progress in this field of study is still limited. Nonetheless, facilitating the adoption of cardiovascular healthy behaviors in the general population seems a promising strategy of intervention to prevent frailty.

## 5. Conclusions

Maintaining a normal weight in adulthood, being physically active, not smoking tobacco, refraining from ultra-processed food and beverages, and avoiding excessive alcohol intake may decrease the risk of frailty several decades later. The framing that frailty is a direct consequence of the normal aging process should be viewed with caution, as we found clear evidence that more vulnerable socioeconomic groups are more likely to become frail.

## Figures and Tables

**Figure 1 healthcare-12-00022-f001:**
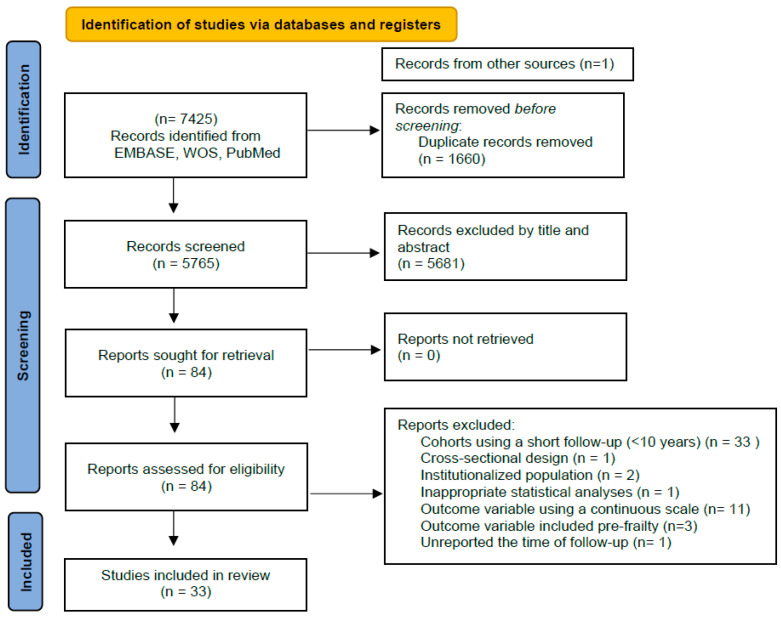
PRISMA 2020 flow diagram showing 33 cohort studies (with at least 10 years of follow-up) of early-life and middle-life risk factors of frailty.

**Table 1 healthcare-12-00022-t001:** Main characteristics of 33 cohort studies included, examining associations between any exposure variable and risk of frailty.

First Author and Year of Publication	Population (Total Sample, Age at Baseline, Sex, Country)	Follow-Up (Mean or Median)	Frailty Definition/ Prevalence of Frailty at Follow-Up	Exposure Variable/Effect Size-Fully Adjusted Model: OR, RR, HR, Beta (95% CI)/Sensitivity Analyses	Covariates of the Fully Adjusted Model
Millar et al. (2022) [17]	*n* = 1701 (55.4% women) Mean 58 years Both sexes USA	12 years	Fried frailty phenotype 13%	Energy-adjusted dietary inflammatory index (E-DII) Per 1-unit increase E-DII: OR = 1.16 (1.07, 1.25) Quartile 4 versus 1: OR = 2.22 (1.37, 3.60) No sensitivity analyses	Age, sex, energy intake, smoking, depressive symptoms, diabetes, cardiovascular disease, and cancer
Strandberg et al. (2018) [18]	*n* = 2360 (92% without chronic diseases or medications at baseline) Mean 49 years Men Finland	30 years	Fried frailty phenotype 10%	Alcohol consumption >196 g per week versus 1–98 g per week OR = 1.61 (1.01, 2.56) No sensitivity analyses	Age, BMI, smoking
Susanto et al. (2018) [19]	*n* = 5462 Median 52 years Women Australia	12 years	FRAIL scale (Abellan van Kan et al.) 7%	Increasing sitting time versus medium sitting time OR = 1.29 (1.03, 1.61) High sitting time versus medium sitting time OR = 1.42 (1.10, 1.84) No sensitivity analyses	Relationship status, education, body mass index, smoking status, alcohol consumption, physical activity, employment, and the presence of arthritis, depression, or hypertension
Sotos-Prieto et al. (2022) [20]	*n* = 121,700 Range 30–55 years Women USA	22 years	FRAIL scale (Morley et al.) 9%	Healthy Heart Score based on smoking, alcohol intake, BMI, physical activity and a diet score that includes five components, namely cereal fibre intake, and consumption of fruits/vegetables, nuts, sugary drinks, and red and processed meats Quintile 5 versus Quintile 1 OR = 5.48 (5.01, 6.00) Excluding individuals with CVD, cancer or diabetes (*n* = 9086), the associations remained similar. Slight attenuation in results from 6-, 10-, and 14-year lag analyses	Age, energy intake, and medication use (aspirin, postmenopausal hormone replacement therapy, diuretics, β-blockers, calcium channel blockers, angiotensin-converting enzyme inhibitors, other blood pressure medication, statins and other cholesterol-lowering drugs, insulin, and oral hypoglycemic medication)
Landré et al. (2020) [21]	*n* = 11,784 (35% Women) Range 61–76 years Both sexes France	26 years	Fried frailty phenotype (modified) Men 5% Women 10%	Overweight versus normal BMI Women OR = 1.79 (1.23, 2.60) Men OR = 1.11 (0.83, 1.48) Obesity versus normal BMI Women OR = 8.18 (5.36, 12.50) Men OR = 4.29 (3.07, 6.01) No sensitivity analyses	Men Age, education, marital status, tobacco and alcohol consumption, diabetes, joint pain, psychological problems, cancer, and cardiovascular disease events Women Age, education, marital status, tobacco and alcohol consumption, diabetes, joint pain, psychological problems, and cancer events
Baranyi et al. (2022) [22]	*n* = 323 (35% Women) Mean 70 years Both sexes UK	Retrospective (early life risk factors)	Frailty index (Mitnitski et al.) Unreported prevalence	Neighborhood social deprivation in childhood (reference category not reported) Men OR = 2.35 (1.40, 4.40) No sensitivity analyses	Covariates unreported
Strandberg et al. (2012) [23]	*n* = 1815 Mean 47 years Men Finland	26 years	Fried frailty phenotype (Modified) 10%	Overweight versus normal BMI OR = 2.06 (1.21, 3.52) Obesity versus normal BMI OR = 5.41 (1.94, 15.10) Composite risk score for Coronary Artery Disease (CAD) per 1-unit increase OR = 1.97 (1.63, 2.39) No sensitivity analyses	Age, weight gain, BMI, smoking, systolic blood pressure, resting heart rate, trigycerides, 1 h post load blood glucose, and composite risk score for CAD
Kheifets et al. (2022) [24]	*n* = 1799 (53% women) Mean age 75 Both sexes Israel	12–14 years	Fried frailty phenotype 14%	Physically inactive versus active OR = 1.71 (0.90, 2.24) No sensitivity analyses	Age, sex, socioeconomic status, heart attack, cardiac insufficiency, other heart disease, stroke, cataract, glaucoma, chronic renal failure, cancer, Alzheimer’s disease, Parkinson’s disease, asthma, other lung disease, diabetes, osteoporosis, dyslipidemia, and hypertension
Landré et al. (2020) [25]	*n* = 12,345 (27% women) Mean age 70 Both sexes France	26 years	Fried frailty phenotype (Modified) 6%	Asthma (at least one report) versus not having asthma OR = 1.50 (1.15, 1.98) No sensitivity analyses	Age, sex, BMI, education, marital status, tobacco consumption, diabetes, joint pain, cancer, cardiac diseases, and mental status
Wennberg et al. (2021) [26]	*n* = 19,341 (61% women) Mean age 72 Both sexes Sweeden	12 years	Hospital Frailty Risk Score (HFRS) Gilbert et al. 27%	Anemia versus normal biomarkers OR = 1.54 (1.38, 1.73) Diabetes versus normal biomarkers OR = 1.59 (1.43, 1.77) Liver enzymes, high versus normal OR = 1.14 (1.01, 1.30) Frailty events during the first year were censored	Age and sex
Bouillon et al. (2013) [27]	*n* = 3895 (27% women) 45–69 years Both sexes UK	10 years	Fried frailty phenotype 3%	Per 1 Standard Deviation increase Framingham CVD risk score OR = 1.42 (1.23, 1.62) Framingham CHD risk score OR = 1.38 (1.20, 1.59) Framingham stroke risk score OR = 1.35 (1.21, 1.51) SCORE (CVD risk score) OR = 1.36 (1.18, 1.56) No sensitivity analyses	Age, sex, and antihypertensive treatment
Pilleron et al. (2017) [28]	*n* = 972 (65% women) Mean 73 years Both sexes France	12 years	Fried frailty phenotype Men 2% Women 4%	Dietary cluster Pasta OR = 2.21 (1.11, 4.40) Dietary cluster Biscuits and Snacking OR = 1.81 (1.17, 2.81) No sensitivity analyses	Marital status, education level, income, multimorbidity (hypertension, diabetes, hypercholesterolemia, angina, cardiac rhythm disorders, cardiac failure, arteritis, myocardial infarction, asthma, Parkinson disease, dyspnea, osteoporosis, and thyroid diseases), BMI, depressive symptomatology, and MMSE (Mini-Mental State Exam)
Haapanen et al. (2018) [29]	*n* = 1078 (56% women) Mean 71 years Both sexes Finland	Retrospective (early life risk factors)	Fried frailty phenotype Men 3% Women 4%	Birth weight (1 kg increase) RRR = 0.36 (0.15, 0.86) Birth length (1 cm increase) RRR = 0.77 (0.66, 0.94) Birth BMI (1 unit increase) RRR = 0.03 (0.001, 0.77) No sensitivity analyses	Age, sex, gestational age, childhood and adulthood SES, adult BMI, smoking, and prevalence of diabetes or hypertension
Haapanen et al. (2018) [30]	*n* = 972 (65% women) Mean 71 years Both sexes Finland	Retrospective (early life risk factors)	Fried frailty phenotype Men 3% Women 4%	Children separated in childhood versus non-separated (only in men) RRR = 5.18 (1.16, 23.17) No sensitivity analyses	Age, sex, gestational age, childhood and adulthood SES, adult BMI, smoking, and prevalence of diabetes or hypertension
Fung et al. (2020) [31]	*n* = 78,366 ≥60 years Women USA	20 years	FRAIL scale (Morley et al.) 16%	Fruits and vegetables, at least 7 portions per day versus less than 3 servings HR = 0.92 (0.85, 0.99) While results for the 8 years lag analysis were weaker, a signal for an inverse association for fruits and vegetables was nevertheless observed	Age, smoking, energy intake, BMI, physical activity, postmenopausal hormone use, aspirin, antihypertensive or lipid lowering medications, diabetes medication, insulin, highest academic degree, census track income data, alcohol, and a modified Alternate Healthy Eating Index that does not include fruits and vegetables
Gil-Salcedo et al. (2020) [32]	*n* = 6357 (29% women) Mean 44 years Both sexes UK	20 years	Fried frailty phenotype 7%	Smoking status (never versus current) HR = 0.68 (0.52, 0.89) Alcohol consumption (moderate versus high) HR = 0.76 (0.59, 0.98) Physical activity (active versus inactive) HR = 0.66 (0.48, 0.88) Fruits and vegetables consumption (at least twice a day) HR = 0.70 (0.53, 0.92) All sensitivity analyses yielded results that were similar to those in the main analyses so that the risk of frailty decreased as the number of healthy behaviors at age 50 increased	Age, sex, ethnicity, marital status, and wave of inclusion, education and occupational position, number of morbidities at age 50 (diabetes, coronary heart disease, stroke, chronic obstructive pulmonary disease, depression, arthritis, cancer, hypertension, and obesity), and all other health behaviors examined
Haapanen et al. (2019) [33]	*n* = 1078 (56% women) Range 67–79 years Both sexes Finland	10 years	Fried frailty phenotype Men 3% Women 4%	Greater BMI gain (>17.5 kg/m^2^) during the period 2–11 yeas was associated with frailty RRR age-adjusted = 2.36 (1.21, 4.63) RRR fully adjusted = 2.07 (0.94, 4.56) No sensitivity analyses	Age, childhood and adulthood SES, adulthood BMI, smoking, hypertension, and diabetes
Orkaby et al. (2022) [34]	*n* = 12,101 ≥60 years Men USA	11 years	Frailty index (Mitnitski et al.) 20%	>60 days of daily Nonsteroideal anti-inflammatory drugs (NSAID) use versus no NSAID use OR = 2.75 (2.29, 3.31) No sensitivity analyses	Propensity score (using comorbidities related to NSAID use, smoking status, and alcohol consumption)
Savela et al. (2013) [35]	*n* = 514 Mean 47 years Men Finland	26 years	Fried frailty phenotype (Modified) Unreported prevalence	High leisure time physical activity (>6 h per week) versus low (<2 h per week) OR = 0.23 (0.08, 0.65) No sensitivity analyses	Age, body mass index, smoking, blood pressure, alcohol consumption, and comorbidity index
Li et al. (2020) [36]	*n* = 6806 (49% women) Mean 69 years Both sexes China	10 years	Fried frailty phenotype (Modified) Unreported prevalence	Neighbourhood quality (highest versus lowest) RR = 0.28 (0.15, 0.52) Educational achievement (at least high school versus illiterate) RR = 0.23 (0.12, 0.44) Paternal education (Literate versus iliterate) RR = 0.74 (0.57, 0.96) Multiple imputation to deal with missing data	Age, sex, residence and marital status), activities of daily living (ADL) disability, and count of comorbidity
Yeung et al. (2021) [37]	*n* = 3702 (51% women Median 72 years Both sexes China	14 years	Fried frailty phenotype Modified frail scale (Morley et al.) 11%	Malnutrition (GLIM criteria) OR = 2.83 (1.47, 5.43) Fried OR = 2.30 (1.06, 4.98) Frail scale No sensitivity analyses	Age, sex, baseline BMI, current smoker, current drinker, live alone, being married, education level, subjective social status, dementia level, depressive symptoms, number of chronic diseases, and physical activity
Brunner et al. (2018) [38]	*n* = 6233 (28% women) Range 45–55 years Both sexes UK	18 years	Fried frailty phenotype 4% Men 3% Women 6%	Current smoking status (versus never) OR = 1.69 (1.27, 2.25) High alcohol consumption (>14 units per week women, >21 units per week men; versus none) OR = 1.54 (1.17, 2.04) Occasional fruit and vegetable consumption (versus daily) OR = 1.29 (1.05, 1.58) Physical inactivity (versus active) OR = 2.63 (2.06, 3.37) Forced expiratory volume (<2.91 L versus >3.58 L) OR = 1.90 (1.36, 2.65) Obesity (versus normal weight) OR = 3.52 (2.62, 4.72) Depressive symptoms (versus none) OR = 1.65 (1.33, 2.03) Hypertension (versus normal blood pressure) OR = 1.39 (1.10, 1.76) HDL cholesterol (>1.59 mmol/L versus <1.25 mmol/L) OR = 1.57 (1.16, 2.12) Interleukin-6 concentration (>1.63 pg/mL versus <1.06 pg/mL) OR = 2.23 (1.59, 3.13) C-reactive protein concentration (>1.37 mg/L versus <0.56 mg/L) OR = 1.94 (1.47, 2.56) Employment grade level (per one unit lower) OR = 1.49 (1.27, 1.75) For employment grade level, sensitivity analysis showed physical activity but not body mass index contributed substantially to the attenuation when removed from the adjustment	Age, sex, time measured since fifth clinic, and ethnicity
Stenholm et al. (2014) [39]	*n* = 1119 Mean 44 years Both sexes Finland	22 years	Fried frailty phenotype 5%	Obesity (versus normal weight) OR = 5.02 (1.89, 13.33) Including only robust participants at the baseline, obesity at baseline predicted development of frailty	Age, sex, education, smoking, alcohol use, physical activity, hypertension, coronary heart disease, other cardiovascular diseases, diabetes, osteoarthritis, inflammatory arthritis, and chronical mental disorder
Walker et al. (2019) [40]	*n* = 5760 (58% women) Range 50–54 years Both sexes USA	24 years	Frailty phenotype by Kucharska-Newton et al. 7%	Inflammation score using fibrinogen, von Willebrand factor, and Factor VIII, and white blood cell count (per 1 standard devation increase) OR = 1.39 (1.18, 1.65) Findings were robust after excluding participants with high inflammatory markers, or clinical stroke or after accounting for bias related to selective attrition	Race, sex, education, socioeconomic status, cognitive status, arthritis, anti-inflammatory medication use, alcohol intake, smoking, and cholesterol markers
Sodhi et al. (2020) [41]	*n* = 1545 (58% women) ≥67 years Both sexes USA	18 years	Fried frailty phenotype (Modified) Unreported prevalence	Pain or discomfort during walking or standing (versus no pain or discomfort) OR = 1.71 (1.41, 2.09) No sensitivity analyses	Age, sex, marital status, education, comorbidity conditions (diabetes, heart attack, stroke, hypertension, cancer, hip fracture, and arthritis), BMI, mini mental state examination, depressive symptoms, and limitations of daily living
Struijk et al. (2022) [42]	*n* = 85,871 ≥60 years Women USA	22 years	FRAIL scale (Morley et al.) 15%	Unprocessed red meat (Per 1 serving per day) OR = 1.08 (1.02, 1.15) Processed red meat (Per 1 serving per day) OR = 1.26 (1.15, 1.39) Physical activity was only in cluded as a covariate in a sensitivity analysis; results showed that including baseline physical activity only marginally lowered the estimates	Age, calendar time, census tract income, education, husband’s education, BMI, smoking status, alcohol intake, energy intake, medication use (aspirin, postmenopausal hormone therapy, diuretics, β-blockers, calcium channel blockers, angiotensin-converting enzyme inhibitors, other blood pressure medication, statins and other cholesterol lowering drugs, insulin, or oral hypoglycaemic medication), consumption of fruits, vegetables, sugar-sweetened beverages, and mutually adjusted for other type of red meat
Dugravot et al. (2020) [43]	*n* = 10,308 (33% women) Range 35–55 years Both sexes UK	24 years (median)	Fried frailty phenotype 27%	Low occupation (versus high) HR = 2.08 (1.85, 2.33) Low education (versus high) HR = 1.08 (0.99, 1.18) Low literacy (versus high) HR = 1.05 (1.01, 1.19) No sensitivity analyses	Sex, race, marital status, and birth cohort Excluded subjects with multimorbidity before 50 years (two or more: diabetes, coronary heart disease, stroke, chronic obstructive pulmonary disease, depression, arthritis, cancer, dementia, and Parkinson’s disease)
Hoogendijk et al. (2018) [44]	*n* = 1509 (52% women) Mean 75 years Both sexes The Netherlands	10 years	Frailty developed by Strawbridge et al. 29%	Low education (versus high) OR = 1.30 (0.73, 2.31) Lowest income (versus highest) OR = 1.90 (1.20, 3.01) Sensitivity analyses (imputation methods) to account for attrition caused by death during follow-up. Same conclusions about patterns of associations were drawn	Sex, year of birth, education, partner status, and income
Yu et al. (2020) [45]	*n* = 694 (50% women) ≥65 years Both sexes China	14 years	Fried frailty phenotype 30%	Subjective social status Low (versus high) OR = 2.34 (1.19, 4.60) No sensitivity analyses	Age, sex, marital status, education, income, hypertension, diabetes, and stroke at baseline, smoking status, alcohol consumption, physical activity, mental health, and cognitive function
Struijk et al. (2020) [46]	*n* = 71,935 Mean 63 years Women USA	22 years	FRAIL scale (Morley et al.) 16%	Sugar sweetened beverages (2 or more servings per day versus never) RR = 1.32 (1.10, 1.57) Artificially sweetened beverages (2 or more servings per day versus never) RR = 1.28 (1.17, 1.39) Orange juice (1–2 servings per day versus never) RR = 0.82 (0.76, 0.87) Non-orange juices (1–2 servings per day versus never) RR = 1.15 (1.03, 1.28) Sensitivity analyses excluding participants with cardiovascular disease, diabetes, cancer, or overweight still showed a significant direct association	Age, calendar time, BMI, smoking status, alcohol intake, energy intake, physical activity, and medication use, overall diet quality, cancer, heart disease, and diabetes diagnosis
Landré et al. (2023) [47]	*n* = 7044 (29% women) Mean 50 years Both sexes UK	21 years	Fried frailty phenotype 7%	Health related quality of life Physical component scores (versus worst quartile) HR = 2.39 (1.85, 3.07) Mental component scores (versus worst quartile) HR = 1.49 (1.15, 1.93) No sensitivity analyses	Sex, occupational position, marital status, ethnicity, and wave at age 50, alcohol consumption, smoking status, physical activity, fruit/vegetable consumption, BMI, and multimorbidity at age 50
Amieva et al. (2022) [48]	*n* = 1531 (% women unreported) Mean 72 years Both sexes France	27 years	Frailty index (Searle et al.) The cut-off of 0.2 points was used to discriminate the frail and robust participants 61%	Social vulnerability index (unreported whether exposure is a continuous or categorical variable) HR = 2.34 (1.08, 5.07) No sensitivity analyses	Age, sex, Instrumental Activies of Daily Living (IADL) disability, comorbidities, and Mini Mental State Examination (MMSE) score
Niederstrasser et al. (2019) [49]	*n* = 7420 (55% women) Mean 67 years Both sexes UK	12 years	Frailty index (Searle et al.) The cut-off higher than 0.25 points was used to discriminate the frail 34%	Vigorous physical activity (versus sedentary) HR = 0.46 (0.36, 0.57) No sensitivity analyses	Age, sex, waist circumference, BMI, income, education, gender, chair raises, smoking, pain, and loneliness

**Table 2 healthcare-12-00022-t002:** Analytical approaches to account for confounding due to pre-existing diseases in 33 studies of risk factors and frailty.

Studies	Covariates Accounted for Diseases in the Maximally Adjusted Model																Morbidity
	CVD	Diabetes	Cancer	Depression	Hypertension	Arthritis	Cataract	Glaucoma	CKD	Alzheimer Cognitive Function	Parkinson	Osteoporosis	Lung Disease	Dyslipidemia	Asthma	Thyroid Disease	(Excluded at Baseline)
Millar et al., 2022 [17]																	
Strandberg et al., 2018 [18]																	
Susanto et al., 2018 [19]																	
Sotos-Prieto et al., 2022 [20]																	
Landré et al., 2020 [21]																	
Baranyi et al., 2022 [22]																	
Strandberg et al., 2012 [23]																	
Kheifets et al., 2022 [24]																	
Landré et al., 2020 [25]																	
Wennberg et al., 2021 [26]																	
Bouillon et al., 2013 [27]																	
Pilleron et al., 2016 [28]																	
Haapanen et al., 2018 [29]																	
Haapaanen et al., 2018 [30]																	
Fung et al., 2020 [31]																	
Gil-Salcedo et al., 2020 [32]																	
Haapaanen et al., 2018 [33]																	
Orkaby et al., 2022 [34]																	
Savela et al., 2013 [35]																	
Li et al., 2020 [36]																	
Yeung et al., 2020 [37]																	
Brunner et al., 2018 [38]																	
Stenhold et al., 2013 [39]																	
Walker et al., 2018 [40]																	
Sodhi et al., 2019 [41]																	
Struijk et al., 2022 [42]																	
Dugravot et al., 2019 [43]																	
Hoogendijk et al., 2017 [44]																	
Yu et al., 2020 [45]																	
Struijk et al., 2020 [46]																	
Niederstrasser et al., 2019 [49]																	
Landré et al., 2023 [47]																	
Amieva et al., 2022 [48]																	

Legend: None; adjusted in the model, excluded participants with morbidities in main analysis, excluded participants with morbidities in sensitivity analysis. CVD: cardiovascular disease; CPK: chronic kidney disease.

**Table 3 healthcare-12-00022-t003:** Score risk of confounding due to pre-existing diseases in 33 studies examining the association between risk factors and frailty.

Studies	Risk of Confounding			
	Age Lower 70 Years	Excluded Diseases	Adjusted Diseases	Score Risk of Confounding
Millar et al., 2022 [17]				1
Strandberg et al., 2018 [18]				2
Susanto et al., 2018 [19]				1
Sotos-Prieto et al., 2022 [20]				1
Landré et al., 2020 [21]				2
Baranyi et al., 2022 [22]				3
Strandberg et al., 2012 [23]				1
Kheifets et al., 2022 [24]				2
Landré et al., 2020 [25]				2
Wennberg et al., 2021 [26]				3
Bouillon et al., 2013 [27]				1
Pilleron et al., 2016 [28]				2
Haapanen et al., 2018 [29]				2
Haapaanen et al., 2018 [30]				2
Fung et al., 2020 [31]				1
Gil-Salcedo et al., 2020 [32]				1
Haapaanen et al., 2018 [33]				2
Orkaby et al., 2022 [34]				2
Savela et al., 2013 [35]				1
Li et al., 2020 [36]				2
Yeung et al., 2020 [37]				1
Brunner et al., 2018 [38]				2
Stenhold et al., 2013 [39]				1
Walker et al., 2018 [40]				0
Sodhi et al., 2019 [41]				2
Struijk et al., 2022 [42]				1
Dugravot et al., 2019 [43]				1
Hoogendijk et al., 2017 [44]				3
Yu et al., 2020 [45]				1
Struijk et al., 2020 [46]				0
Landré et al., 2023 [47]				1
Amieva et al., 2022 [48]				2
Niederstrasser et al., 2019 [49]				2

Legend: red color: NO; green color: YES; high risk of confounding: score 2–3; low risk of confounding: score 0–1.

## Data Availability

No new data were created in this systematic review. Readers may contact with corresponding author to clarify any doubt regarding the methodology used.

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
