# Peer review of "Understanding the Causes of Frailty Using a Life-Course Perspective: A Systematic Review"

_healthcare, 2023, doi:10.3390/healthcare12010022_

Round 1
Reviewer 1 Report
Comments and Suggestions for Authors
The aim of this article was to identify risk factors associated with frailty in early life and adulthood. The article systematically collected evidence and made scientific analysis. The article is of great value to the study of geriatric health. But I think there are still some limitations of the article.
1. Form The age of the baseline form table one, there are very different age range of the studies. So, the readers can't clearly see early life and adulthood influence form the table.
2. Some studies the women are far less than man, this may cause bias.
3. Given the differences in samples and data collection, the strength of the scientific evidence provided by the studies varied, and whether the authors took this into account.
Author Response
REVIEWER 1
The aim of this article was to identify risk factors associated with frailty in early life and adulthood. The article systematically collected evidence and made scientific analysis. The article is of great value to the study of geriatric health. But I think there are still some limitations of the article.
Comment
- From the age of the baseline from table one, there are very different age range of the studies. So, the readers can't clearly see early life and adulthood influence from the table.
Response
Thanks for your comment. To help the readers to identify what studies evaluated early life risk factors of frailty, in Table 1 we have added information (see the second column) about what studies evaluated early life risk factors.
Comment
- Some studies the women are far less than man, this may cause bias.
Response
Table 1 clearly shows that most studies included samples with both sexes (n=24). From the nine studies that only evaluated one sex, five were in women whereas four were conducted in men. In general, we can conclude that participants of both sexes have contributed in the literature of risk factors of frailty.
Comment
- Given the differences in samples and data collection, the strength of the scientific evidence provided by the studies varied, and whether the authors took this into account.
Response
As we clearly say, a meta-analysis would be helpful to summarize the synthesis of the evidence obtained from different sample sizes. However, we could not run meta-analysis due to the few number of studies found for each exposure variable.
Reviewer 2 Report
Comments and Suggestions for Authors
Firstly, I would like to acknowledge and thank the authors for the efforts put into this research, and would like to address some questions in order to clarify and eventually improve the quality of the existing manuscript in order to be published and made available to the scientific community.
Study goals and title
Article main subject is interesting, and is of major relevance to the area of public health.
According to the abstract, this systematic review aims to identify early life and adulthood risk factors associated with frailty. Considering that aim, the title does not seem to match this aim. Moreover, the aim described in the Introduction section includes another variable: “(…) in older adults.” Coherence should be considered when describing the aims of the study throughout the manuscript and title. Therefore, I suggest using the same definition of goals for the study and readjusting the title of the paper.
Please consider following the PICO in order to clarify and to maintain consistency along the manuscript (the same for the methods section).
Background / Introduction
Background defines clearly some of the concepts that are addressed in the study. However, considering the two definitions of “frailty” mentioned, it´s important (here on in the methods section) clearly indicate which one will serve as a reference in the present study.
I believe that the study by Taylor (reference no.5 in line 41) it´s not a systematic review. It is a review, however, it´s not a systematic review, therefore should be named that way: review (and not systematic review).
Not questioning the strengths of the proposed study, I believe that with regards to the justification of the study, there seems to be several arguments related to the more adequate public health studies´ methods however, the rationale for the review is not presented in the context of existing knowledge, especially in terms of previously published reviews and or metanalysis on the subject. There should be clearly identified, in light of the published systematic reviews and meta-analysis on the subject (and not only one mentioned in line 47) what gap in knowledge will this work give its contribution.
Methods
In line 72 there is a reference to “a list of keywords is included in Table S1”, however I believe that that Supplementary table was not provided and its of major importance to allow study replication. Please provide the named table otherwise consider including that information in the content of this section.
For the same reason, more detail should be included in the description of eligibility criteria, information sources, and search strategy, to allow study replication. Could you please provide information on the following?
What excluding criteria other than a follow-up lower than 10 years (line 77) were considered? There are other excluding criteria described in Figure 1. If not described in the methods section one might think they were not set a priori, as they should (and is probably not the case). Please, describe clearly all criteria used for exclusion.
What filters and limits were used?
Was there any language restriction?
Any type of studies excluded? Why? How were previous systematic reviews and meta-analysis considered to this review, and why?
Eligibility criteria should be clear in terms of the population characteristics (ages, sex, health or other issues or diseases).
Were reference lists or other sources considered? Please describe them in methods, as one can understand that maybe there were from looking at Figure 1. If yes (as described in Figure 1) please indicate studies “from other sources” in the correct box according to the PRISMA 2020 flow diagram.
Results
Please see notes indicated for section “methods” that refer to the Study flow diagram.
As a suggestion, Table 2 could be filled with information that don’t rely on color to be interpreted as (probably) printed version of the manuscript will be black and white. Also, could help with readability too.
The same suggestion applies to Table 3.
Studies should indicate the reference number in both tables for better correspondence to the text presented in the results and discussion sections.
Discussion
Discussion section seems very reductive and appear to raise more questions than explaining what was found. It should help readers interpret the findings and, most of the time, it doesn´t. It also should provide information relevant to future studies and/or interventions, and it doesn´t.
In my opinion, the beginning of this section should indicate the goal of the study and not state the findings of the study. As an example: statements in lines 170-173 are just that – statements - and do not explore or explain what are the implications and what can be done in the future.
Also, the text states recurrently that topics/findings are “partially known” or “remain unclear” which again, does not contribute to clarification and the expected information on the studied subject.
I would like authors to analyse in details what studies agree/disagree on results and is that related to their methods? Do studies with same methods have similar results? Do different methods obtain the same results? Do methods affect results when comparing studies? Is there any interesting fact that can be discussed considering the different population characteristics of the studies (by country, sex, etc.)?
Considering that you chose this as one of your main conclusions your work, what could you add to what you mentioned in lines 194-196: “So far, the mechanisms linking socioeconomic factors with frailty remain unexplored, and multiple factors may be involved (beyond the proven benefits of physical activity or healthy diets).”? There seems to be no further information to explain exactly what is there to be considered regarding this observation.
Could you please indicate what information would you like to add to your statement: “future studies should adopt (at least) the most common definitions to allow a future synthesis of the scientific evidence” in lines 197-199? What is your “specific” advice for future studies since? What do you think that are the most common definitions?
Could you please explain what´s your understanding of “with appropriate methodologies of assessment” referred in line 203?
No comments on the conclusions section.
I would like authors to consider my suggestions and clarify or correct whenever asked in order to improve the quality of the final work.
Good job.
Author Response
REVIEWER 2
Comment
Firstly, I would like to acknowledge and thank the authors for the efforts put into this research, and would like to address some questions in order to clarify and eventually improve the quality of the existing manuscript in order to be published and made available to the scientific community.
Response
Thanks for your kind words and for help us to improve the quality of the manuscript.
Comment
Study goals and title
Article main subject is interesting, and is of major relevance to the area of public health.
According to the abstract, this systematic review aims to identify early life and adulthood risk factors associated with frailty. Considering that aim, the title does not seem to match this aim. Moreover, the aim described in the Introduction section includes another variable: “(…) in older adults.” Coherence should be considered when describing the aims of the study throughout the manuscript and title. Therefore, I suggest using the same definition of goals for the study and readjusting the title of the paper.
Response
We agree with the reviewer that the description of the population examined in our review might be confusing. However, we think that the title and descriptions in the text are accurate scientifically. For example, when we refer to life-course perspective we mean the whole life of individuals. Therefore, we were interested to evaluate early life risk factors of future frailty (for example, characteristics in babies, children or adolescents), as well as factors during middle adulthood. Our study is novel, because previous systematic reviews included information of risk factors of frailty that were commonly evaluated in older adults.
Comment
Please consider following the PICO in order to clarify and to maintain consistency along the manuscript (the same for the methods section).
Response
Thanks for your wise comment. Following your advice, we have rephrased section 2.1 using the PICO model.
Comment
Background / Introduction
Background defines clearly some of the concepts that are addressed in the study. However, considering the two definitions of “frailty” mentioned, it´s important (here on in the methods section) clearly indicate which one will serve as a reference in the present study.
Response
The aim of our study was not to evaluate the validity of different definitions of frailty. Thus, we did not exclude studies based on type of definition used; or we do not express our view on the merits of different proposals of frailty definitions. We think that in the future, researchers should compare different definitions and make a consensus definition to be commonly used among epidemiologists.
Comment
I believe that the study by Taylor (reference no.5 in line 41) it´s not a systematic review. It is a review, however, it´s not a systematic review, therefore should be named that way: review (and not systematic review).
Response
Thanks for your comment. We agree with the changed suggested. In the new version, in the introduction section reference 5 is now mentioned as a narrative review.
Comment
Not questioning the strengths of the proposed study, I believe that with regards to the justification of the study, there seems to be several arguments related to the more adequate public health studies´ methods however, the rationale for the review is not presented in the context of existing knowledge, especially in terms of previously published reviews and or metanalysis on the subject. There should be clearly identified, in light of the published systematic reviews and meta-analysis on the subject (and not only one mentioned in line 47) what gap in knowledge will this work give its contribution.
Response
Following the reviewer’s suggestion, we have added a short paragraph (line 63) that may help the readers to understand the rationale to conduct our systematic review.
Comment
Methods
In line 72 there is a reference to “a list of keywords is included in Table S1”, however I believe that that Supplementary table was not provided and its of major importance to allow study replication. Please provide the named table otherwise consider including that information in the content of this section.
Response
In the new version, the search terms used in the systematic review are now included within Table S1. Using this list of words, authors may replicate our search strategy.
Comment
For the same reason, more detail should be included in the description of eligibility criteria, information sources, and search strategy, to allow study replication. Could you please provide information on the following? What excluding criteria other than a follow-up lower than 10 years (line 77) were considered? There are other excluding criteria described in Figure 1. If not described in the methods section one might think they were not set a priori, as they should (and is probably not the case). Please, describe clearly all criteria used for exclusion. What filters and limits were used? Was there any language restriction? Any type of studies excluded? Why? How were previous systematic reviews and meta-analysis considered to this review, and why? Eligibility criteria should be clear in terms of the population characteristics (ages, sex, health or other issues or diseases). Were reference lists or other sources considered? Please describe them in methods, as one can understand that maybe there were from looking at Figure 1. If yes (as described in Figure 1) please indicate studies “from other sources” in the correct box according to the PRISMA 2020 flow diagram.
Response
Following the reviewer’s suggestion we have added more information in the text about the eligibility criteria to inform the reader (see lines 84-91); and we have edited the PRISMA 2020 flow diagram.
Comment
Results
Please see notes indicated for section “methods” that refer to the Study flow diagram. As a suggestion, Table 2 could be filled with information that don’t rely on color to be interpreted as (probably) printed version of the manuscript will be black and white. Also, could help with readability too. The same suggestion applies to Table 3. Studies should indicate the reference number in both tables for better correspondence to the text presented in the results and discussion sections.
Response
Here, we prefer to maintain the format sent using several colors. One the authors (Rezende L) previously published this exact same format in reference 14.
Comment
Discussion
Discussion section seems very reductive and appear to raise more questions than explaining what was found. It should help readers interpret the findings and, most of the time, it doesn´t. It also should provide information relevant to future studies and/or interventions, and it doesn´t.
In my opinion, the beginning of this section should indicate the goal of the study and not state the findings of the study. As an example: statements in lines 170-173 are just that – statements - and do not explore or explain what are the implications and what can be done in the future.
Response
Done. In the new version, we start the discussion section explaining the main goal of the systematic review and briefly summarize the main findings. In addition, we have added a new reference to state that interventions late in life to postpone or delay frailty may not be effective.
Comment
Also, the text states recurrently that topics/findings are “partially known” or “remain unclear” which again, does not contribute to clarification and the expected information on the studied subject.
I would like authors to analyse in details what studies agree/disagree on results and is that related to their methods? Do studies with same methods have similar results? Do different methods obtain the same results? Do methods affect results when comparing studies? Is there any interesting fact that can be discussed considering the different population characteristics of the studies (by country, sex, etc.)?
Considering that you chose this as one of your main conclusions your work, what could you add to what you mentioned in lines 194-196: “So far, the mechanisms linking socioeconomic factors with frailty remain unexplored, and multiple factors may be involved (beyond the proven benefits of physical activity or healthy diets).”? There seems to be no further information to explain exactly what is there to be considered regarding this observation.
Response
We think that it is difficult to synthesize the main findings using a systematic review design without meta-analysis because you cannot discuss how the overall score is consistent among different studies (heterogeneity). Therefore, we preferred to short the discussion and only highlight the main trends found. However, in the new version we have expanded the discussion adding more relevant information and based on the reviewer suggestions.
Comment
Could you please indicate what information would you like to add to your statement: “future studies should adopt (at least) the most common definitions to allow a future synthesis of the scientific evidence” in lines 197-199? What is your “specific” advice for future studies since? What do you think that are the most common definitions?
Response
Thanks for this suggestion. We have amended this paragraph adding the term “frailty” to improve the quality of the writing. Regarding frailty definitions see changes in line 241 when we name Fried definition.
Comment
Could you please explain what´s your understanding of “with appropriate methodologies of assessment” referred in line 203?
Response
Thanks. We provide further details in the new version (see 255-257).
----------------------------------------------------------------------------------------------
No comments on the conclusions section.
I would like authors to consider my suggestions and clarify or correct whenever asked in order to improve the quality of the final work.
Response
Thanks for your excellent tips.